# Meta-Reinforcement Learning with Self-Modifying Networks

**Mathieu Chalvidal**
Artificial and Natural Intelligence Toulouse Institute
Universite de Toulouse, France
`mathieu_chalvid@brown.edu`

**Thomas Serre**
Carney Institute for Brain Science
Brown University, U.S.
`thomas_serre@brown.edu`

**Rufin VanRullen**
Centre de Recherche Cerveau & Cognition
CNRS, Universite de Toulouse, France
`rufin.vanrullen@cnrs.fr`

## Abstract

Deep Reinforcement Learning has demonstrated the potential of neural networks tuned with gradient descent for solving complex tasks in well-delimited environments. However, these neural systems are slow learners producing specialized agents with no mechanism to continue learning beyond their training curriculum. On the contrary, biological synaptic plasticity is persistent and manifold, and has been hypothesized to play a key role in executive functions such as working memory and cognitive flexibility, potentially supporting more efficient and generic learning abilities. Inspired by this, we propose to build networks with dynamic weights, able to continually perform self-reflexive modification as a function of their current synaptic state and action-reward feedback, rather than a fixed network configuration. The resulting model, MetODS (for Meta-Optimized Dynamical Synapses) is a broadly applicable meta-reinforcement learning system able to learn efficient and powerful control rules in the agent policy space. A single layer with dynamic synapses can perform one-shot learning, generalizes navigation principles to unseen environments and manifests a strong ability to learn adaptive motor policies.

## 1 Introduction

The algorithmic shift from hand-designed to learned features characterizing modern Deep Learning has been transformative for Reinforcement Learning (RL), allowing to solve complex problems ranging from video games [1, 2] to multiplayer contests [3] or motor control [4, 5]. Yet, "deep" RL has mostly produced specialized agents unable to cope with rapid contextual changes or tasks with novel or compositional structure [6–8]. The vast majority of models have relied on gradient-based optimization to learn static network parameters adjusted during a predefined curriculum, arguably preventing the emergence of online adaptivity. A potential solution to this challenge is to *meta-learn* [9–11] computational mechanisms able to rapidly capture a task structure and automatically operate complex feedback control: Meta-Reinforcement Learning constitutes a promising direction to build more adaptive artificial systems [12] and identify key neuroscience mechanisms that endow humans with their versatile learning abilities [13].

In this work, we draw inspiration from biological fast synaptic plasticity, hypothesized to orchestrate flexible cognitive functions according to context-dependent rules [14–17]. By tuning neuronal selectivity at fast time scales – from fast neural signaling (milliseconds) to experience-based learning

36th Conference on Neural Information Processing Systems (NeurIPS 2022).

(seconds and beyond) – fast plasticity can support in principle many cognitive faculties including motor and executive control. From a dynamical system perspective, fast plasticity can serve as an efficient mechanism for information storage and manipulation and has led to modern theories of working memory [18–22]. Despite the fact that the magnitude of the synaptic gain variations may be small, such modifications are capable of profoundly altering the network transfer function [23] and constitute a plausible mechanism for rapidly converting reward and choice history into tuned neural functions [24]. From a machine learning perspective, despite having a long history [25–29], fast weights have most often been investigated in conjunction with recurrent neural activations [30–33] and rarely as a function of an external reward signal or of the current synaptic state itself. Recently, new proposals have shown that models with dynamic modulation related to fast weights could yield powerful meta-reinforcement learners [34–36]. In this work, we explore an original self-referential update rule that allows the model to form synaptic updates conditionally on information present in its own synaptic memory. Additionally, environmental reward is injected continually in the model as a rich feedback signal to drive the weight dynamics. These features endow our model with a unique recursive control scheme, that support the emergence of a self-contained reinforcement learning program.

**Contribution:** We demonstrate that a neural network trained to continually self-modify its weights as a function of sensory information and its own synaptic state can produce a powerful reinforcement learning program. The resulting general-purpose meta-RL agent called 'MetODS' (for Meta-Optimized Dynamical Synapses) is theoretically presented as a model-free approach performing stochastic feedback control in the policy space. In our experimental evaluation, we investigate the reinforcement learning strategies implemented by the model and demonstrate that a single layer with lightweight parametrization can implement a wide spectrum of cognitive functions, from one-shot learning to continuous motor-control. We hope that MetODS inspires more works around self-optimizing neural networks.

The remainder of the paper is organised as follows: In Section 2 we introduce our mathematical formulation of the meta-RL problem, which motivates MetODS computational principles presented in Section 3. In Section 4 we review previous approaches of meta-reinforcement learning and we discuss other models of artificial fast plasticity and their relation to associative memory. In Section 5 we report experimental results in multiple contexts. Finally, in Section 6 we summarise the main advantages of MetODS and outline future work directions.

## 2   Background

### 2.1   Notation

Throughout, we refer to "tasks" as Markov decision processes (MDP) defined by the following tuple $\tau = (\mathcal{S}, \mathcal{A}, \mathcal{P}, r, \rho_0)$, where $\mathcal{S}$ and $\mathcal{A}$ are respectively the state and action sets, $\mathcal{P} : \mathcal{S} \times \mathcal{A} \times \mathcal{S} \mapsto [0, 1]$ refers to the state transition distribution measure associating a probability to each tuple (state, action, new state), $r : \mathcal{A} \times \mathcal{S} \mapsto \mathbb{R}$ is a bounded reward function and $\rho_0$ is the initial state distribution. For simplicity, we consider finite-horizon MDP with $T$ time-steps although our discussion can be extended to the infinite horizon case as well as partially observed MDP. We further specify notation when needed by subscripting symbols with the corresponding task $\tau$ or time-step $t$.

### 2.2   Meta-Reinforcement learning as an optimal transport problem:

*Meta-Reinforcement learning* considers the problem of building a program that generates a distribution $\mu_\pi$ of policies $\pi \in \Pi$ that are "adapted" for a distribution $\mu_\tau$ of task $\tau \in \mathbb{T}$ with respect to a given metric $\mathcal{R}$. For instance, $\mathcal{R}(\tau, \pi)$ can be the expected cumulative reward for a task $\tau$ and policy $\pi$, i.e where state transitions are governed by $s_{t+1} \sim \mathcal{P}_\tau(.|s_t, a_t)$, actions are sampled according to the policy $\pi$: $a_t \sim \pi$ and initial state $s_0$ follows the distribution $\rho_{0,\tau}$. Provided the existence of an optimal policy $\pi^*$ for any task $\tau \in \mathbb{T}$, we can define the distribution measure $\mu_{\pi^*}$ of these policies over $\Pi$. Arguably, an ideal system aims at associating to any task $\tau$ its optimal policy $\pi^*$, i.e, finding the transport plan $\gamma$ in the space $\Gamma(\mu_\mathbb{T}, \mu_{\pi^*})$ of coupling distributions with marginals $\mu_\mathbb{T}$ and $\mu_{\pi^*}$ that maximizes $\mathcal{R}$.

$$\max_{\gamma \in \Gamma(\mu_\mathbb{T}, \mu_{\pi^*})} \mathbb{E}_{(\tau, \pi) \sim \gamma} \left[ \mathcal{R}(\tau, \pi) \right] \quad \text{where} \quad \mathcal{R}(\tau, \pi) = \mathbb{E}_{\pi, \mathcal{P}_\tau} \left[ \sum_{t=0}^{T} r_\tau(a_t, s_t) \right] \tag{1}$$

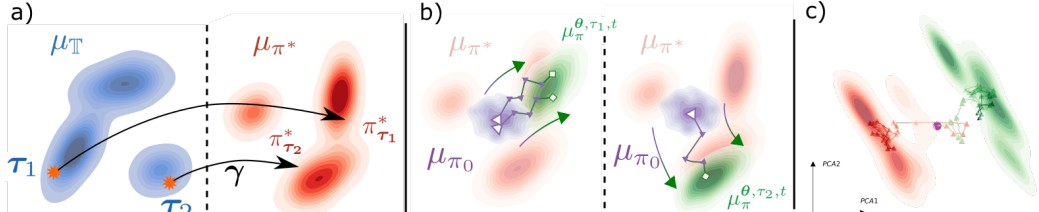

**Figure 1: Meta-Reinforcement Learning as a transport problem and MetODS synaptic adaptation**: **a)** Associating any task $\tau$ in $\mathbb{T}$ to its optimal policy $\pi_\tau^*$ in $\Pi$ can be regarded as finding an optimal transport plan from $\mu_\mathbb{T}$ to $\mu_{\pi^*}$ with respect to the cost $-\mathcal{R}$. Finding this transport plan is generally an intractable problem. **b)** Meta-RL approximates a solution by defining a stochastic flow in the policy space $\Pi$ conditioned by the current task $\tau$ and driving a prior distribution $\mu_{\pi_0}$ of policies $\pi_0$ towards a distribution $\mu_\pi^{\theta,\tau,t}$ of policies with high score $\mathcal{R}$. **c)** Density and mean trajectories of our model dynamic weights principal components over several episodes of the Harlow task (see section 5.1) reveal this policy specialization. Two modes colored with respect to whether the agent initial guess was good or bad emerge, corresponding to two different policies to solve the task.

Most generally, problem (1) is intractable, since $\mu_{\pi^*}$ is unknow or has no explicit form. Instead, previous approaches optimize a surrogate problem, by defining a parametric specialization procedure which builds for any task $\tau$, a sequence $(\pi_t)_t$ of improving policies (see Fig. 1). Defining $\boldsymbol{\theta}$ some meta-parameters governing the evolution of the sequences $(\pi_t)_t$ and $\mu_\pi^{\boldsymbol{\theta},\tau,t}$ the distribution measure of the policy $\pi_t$ after learning task $\tau$ during some period $t$, the optimization problem amounts to finding the meta-parameters $\boldsymbol{\theta}$ that best adapt $\pi_t \sim \mu_\pi^{\boldsymbol{\theta},\tau,t}$ over the task distribution.

$$\max_{\boldsymbol{\theta}} \quad \mathbb{E}_{\tau \sim \mu_\mathbb{T}} \left[ \mathbb{E}_{\pi \sim \mu_\pi^{\boldsymbol{\theta},\tau,t}} \left[ \mathcal{R}(\tau, \pi) \right] \right] \tag{2}$$

Equation (2) cast meta-Reinforcement learning as the problem of building automatic policy control in $\Pi$. We discuss below desirable properties of such control for which we show that our proposed meta-learnt synaptic rule has a good potential.

$\diamond$ **Efficiency:** How "fast" the distribution $\mu_\pi^{\boldsymbol{\theta},\tau,t}$ is transported towards a distribution of high-performing policies. Arguably, an efficient learning mechanism should require few interaction steps $t$ with the environment to identify the task rule and adapt its policy accordingly. This can be seen as minimizing the agent's cumulative regret during learning [37]. This notion is also connected to the policy exploration-exploitation trade-off [38, 39], where the agent learning program should foster rapid discovery of policy improvements while retaining performance gains acquired through exploration. We show that our learnt synaptic update rule is able to change the agent transfer function drastically in a few updates, and settle to a policy when task structure has been identified, thus supporting one-shot learning of a task-contingent association rule in the Harlow task, adapting a motor policy in a few steps or exploring original environments quickly in the Maze experiment.

$\diamond$ **Capacity:** That property defines the learner sensitivity to particular task structures and its ability to convert them into precise states in the policy space, which determines the achievable level of performance for a distribution of tasks $\mu_\tau$. This notion is linked to the sensitivity of the learner, i.e how the agent captures and retains statistics and structures describing a task. Precedent work has shown that memory-based Meta-RL models operate a Bayesian task regression problem [40], and further fostering task identification through diverse regularizations [41, 42, 39] benefited performance of the optimized learning strategy. Because our mechanism is continual, it allows for constant tracking of the environment information and policy update similar to memory-based models. We particularly test this property in the maze experiment in Section 5, showing that tuned online synaptic updates obtain the best capacity under systematic variation of the environment.

$\diamond$ **Generality:** We refer here to the overall ability of the meta-learnt policy flow to drive $\mu_\pi^{\boldsymbol{\theta},\tau,t}$ towards high-performing policy regions for a diverse set of tasks (*generic trainability*) but also to how general is the resulting reinforcement learning program and how well it transfers to tasks unseen during training (*transferability*). In the former case. since the proposed synaptic mechanism is model-free, it allows for tackling diverse types of policy learning, from navigation to motor control. Arguably, to build reinforcing agents that learn in open situations, we should strive for generic and efficient computational mechanisms rather than learnt heuristics. *Transferability* corresponds to the ability of the meta-learned policy flow to generally yield improving updates even in unseen policy regions

of the space $\Pi$ or conditioned by unseen task properties: new states, actions and transitions, new reward profile. etc. We show in a motor-control experiment using the Meta-World benchmark that meta-tuned synaptic updates are a potential candidate to produce a more systematic learner agnostic to environment setting and reward profile. The generality property remains the hardest for current meta-RL approaches, demonstrating the importance of building more stable and invariant control principles.

## 3  MetODS: Meta-Optimized Dynamical Synapses

### 3.1  Learning reflexive weights updates

What if a neural agent could adjust the rule driving the evolution of $\mu_\pi^{\theta,\tau,t}$ based on its own knowledge? In this work, we test a neural computational model that learns to compress experience of a task $\tau$ into its dynamic weights $W_t$.

$$\forall t \leq T, \quad \pi(\boldsymbol{a}|\boldsymbol{s}, \boldsymbol{W}_t) \sim \mu_\pi^{\boldsymbol{\theta},\tau,t} \qquad (3)$$

Contrary to gradient-based rules whose analytical expression is a static and predefined function of activations error ($\Delta(\boldsymbol{W}_t) = \frac{\partial \boldsymbol{a}}{\partial \boldsymbol{W}} \cdot \frac{\partial \mathcal{L}}{\partial \boldsymbol{a}}$), we define an update rule that directly depends on the current weight state through a parameterized non-linear recursive scheme, making the expression of the update much more sensitive to weight content itself.

$$\forall t \leq T, \quad \Delta(\boldsymbol{W}_t) = \mathcal{F}_\theta(\boldsymbol{W}_t)\Big|_{\boldsymbol{W}=\boldsymbol{W}_t} \qquad (4)$$

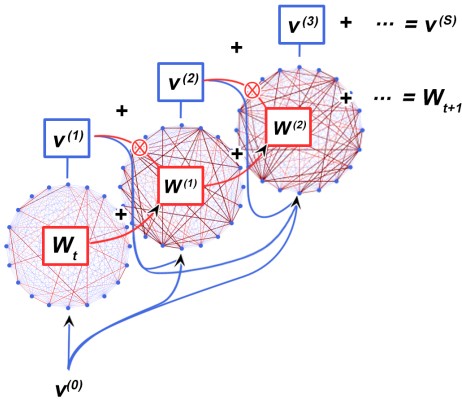

**Figure 2:** A layer of MetODS updates its weights by recursive applications of **read** and **write** operations based on neural activations $\boldsymbol{v}^{(s)}$ and synaptic traces $\boldsymbol{W}^{(s)}$.

This self-reflexive evolution of the synaptic configuration $\boldsymbol{W}_t$ allows for online variation of the learning rule during adaptation for a given task $\tau$ and opens an original repertoire of dynamics for the policy evolution $\mu_\pi^{\boldsymbol{\theta},\tau,t}$. Specifically, at every time-step $t$, network inference and learning consists in recursive applications of read and write operations that we define below.

$\diamond$ **Read-write operations:** The core mechanism consists in two simple operations that respectively linearly project neurons activation $\boldsymbol{v}$ through the dynamic weights $\boldsymbol{W}$ followed by an element-wise non-linearity, and build a local weight update with element-wise weighting $\boldsymbol{\alpha}$:

$$\begin{cases} \phi(\boldsymbol{W},\boldsymbol{v}) = \sigma(\boldsymbol{W}.\boldsymbol{v}) & read \\ \psi(\boldsymbol{v}) = \boldsymbol{\alpha} \odot \boldsymbol{v} \otimes \boldsymbol{v} & write \end{cases} \qquad (5)$$

Here, $\boldsymbol{\alpha}$ is a matrix of $\mathbb{R}^{N \times N}$, $\otimes$ denotes the outer-product, $\odot$ is the element-wise multiplication, $\sigma$ is a non-linear activation function. The read and write operations are motivated by biological synaptic computation: Reading corresponds to the non-linear response of the neuron population to a specific activation pattern. Writing consists in an outer product that emulates a local Hebbian rule between neurons. The element-wise weighting $\boldsymbol{\alpha}$ allows locally tuning synaptic plasticity at every connection consistent with biology [43, 44] which generates a matrix update with potentially more than rank one as in the classic hebbian rule.

$\diamond$ **Recursive update:** While a single iteration of these two operations can only retrieve a similar activation pattern (reading) or add unfiltered external information into weights (writing), recursively applying these operations offers a much more potent computational mechanism that mix information between current neural activation and previous iterates. Starting from an initial activation pattern $\boldsymbol{v}^{(0)}$ and previous weight state $\boldsymbol{W}^{(0)} = \boldsymbol{W}_{t-1}$, the model recursively applies equations (5) for $s \in [1, S]$ on $\boldsymbol{v}^{(s)}$ and $\boldsymbol{W}^{(s)}$ such that:

$$\text{for } s \in [1, S] \quad : \quad \begin{cases} \boldsymbol{v}^{(s)} = \sum_{l=0}^{s-1} \boldsymbol{\kappa}_s^{(l)} \boldsymbol{v}^{(l)} + \boldsymbol{\kappa}_s^{(s)} \phi(\boldsymbol{W}^{(s-1)}, \boldsymbol{v}^{(s-1)}) \\ \boldsymbol{W}^{(s)} = \sum_{l=0}^{s} \boldsymbol{\beta}_s^{(l)} \boldsymbol{W}^{(l)} + \boldsymbol{\beta}_s^{(s)} \psi(\boldsymbol{v}^{(s-1)}) \end{cases} \qquad (6)$$

Parameters $\boldsymbol{\kappa}_s^{(l)}$ and $\boldsymbol{\beta}_s^{(l)}$ are scalar values learnt along with plasticity parameters $\boldsymbol{\alpha}$, and correspond to delayed contributions of previous patterns and synaptic states to the current operations. This is motivated by biological evidence of complex cascades of temporal modulation mechanisms over synaptic efficacy [45, 46]. Finally, $(\boldsymbol{v}^{(S)}, \boldsymbol{W}^{(S)})$ are respectively used as activation for the next layer, and as the new synaptic state $\boldsymbol{W}_t$.

$\diamond$ **Computational interpretation:** We note that if S=1 in equation (6), the operation boils down to a simple hebbian update with a synapse-specific weighting $\boldsymbol{\alpha}^{i,j}$. This perspective makes MetODS an original form of modern Hopfield network [47] with hetero-associative memory that can dynamically access and edit stored representations driven by observations, rewards and actions. While pattern retrieval from Hopfield networks has a dense litterature, our recursive scheme is an original proposal to learn automatic updates able to articulate representations across timesteps. We believe that this mechanism is particularly benefitial for meta-RL to filter external information with respect to past experience. The promising results shown in our experimental section suggest that such learnt updates can generate useful self-modifications to sequentially adapt to incoming information at runtime.

In this work, we test a single dynamic layer for MetODS and leave the extension of the synaptic plasticity to the full network for future work. In order for the model to learn a credit assignment strategy, state transition and previous reward information $[\boldsymbol{s}_t, \boldsymbol{a}_{t-1}, \boldsymbol{r}_{t-1}]$ are embedded into a vector $\boldsymbol{v}_t$ by a feedforward map $\boldsymbol{f}$ as in previous meta-RL approaches [48, 49]. Action and advantage estimate are read-out by a feedforward

---

**Algorithm 1:** MetODS synaptic learning

1: **Require:** $\boldsymbol{\theta} = [\boldsymbol{f}, \boldsymbol{g}, \boldsymbol{\alpha}, \boldsymbol{\kappa}, \boldsymbol{\beta}]$ and $\boldsymbol{W}_0$
2: **for** $1 \leq t \leq T$ **do**
3: $\quad \boldsymbol{v}^{(0)} \leftarrow \boldsymbol{f}(\boldsymbol{s}_t, \boldsymbol{a}_{t-1}, \boldsymbol{r}_{t-1})$
4: $\quad \boldsymbol{W}^{(0)} \leftarrow \boldsymbol{W}_{t-1}$
5: $\quad$ **for** $1 \leq s \leq S$ **do**
6: $\quad\quad \boldsymbol{v}^{(s)} \leftarrow \sum_{l=0}^{s-1} \boldsymbol{\kappa}_s^{(l)} \boldsymbol{v}^{(l)} + \boldsymbol{\kappa}_s^{(s)} \sigma(\boldsymbol{W}^{(s-1)}.\boldsymbol{v}^{(s-1)})$
7: $\quad\quad \boldsymbol{W}^{(s)} \leftarrow \sum_{l=0}^{s-1} \boldsymbol{\beta}_s^{(l)} \boldsymbol{W}^{(l)} + \boldsymbol{\beta}_s^{(s)}(\boldsymbol{\alpha} \odot \boldsymbol{v}^{(s-1)} \otimes \boldsymbol{v}^{(s-1)})$
8: $\quad$ **end for**
9: $\quad \boldsymbol{a}_t, \boldsymbol{v}_t \leftarrow \boldsymbol{g}(\boldsymbol{v}^{(s)})$
10: $\quad \boldsymbol{W}_t \leftarrow \boldsymbol{W}^{(s)}$
11: **end for**

---

policy map $\boldsymbol{g}$. We meta-learn the plasticity and update coefficients, as well as the embedding and read-out function altogether: $\boldsymbol{\theta} = [\boldsymbol{f}, \boldsymbol{g}, \boldsymbol{\alpha}, \boldsymbol{\kappa}, \boldsymbol{\beta}]$. Additionally, the initial synaptic configuration $\boldsymbol{W}_0$ can be learnt, fixed a priori or sampled from a specified distribution.

# 4 Related work

$\diamond$ **Meta-Reinforcement learning** has recently flourished into several different approaches aiming at learning high-level strategies for capturing task rules and structures. A direct line of work consists in automatically meta-learning components or parameters of the RL arsenal to improve over heuristic settings [50–52]. Orthogonally, work building on the Turing-completeness of recurrent neural networks has shown that simple recurrent neural networks can be trained to store past information in their persistent activity state to inform current decision, in such a way that the network implements a form of reinforcement learning over each episode [53, 48, 54]. It is believed that vanilla recurrent networks alone are not sufficient to meta-learn the efficient forms of episodic control found in biological agents [55, 13]. Hence additional work has tried to enhance the system with a better episodic memory model [56–58] or by modeling a policy as an attention module over an explicitly stored set of past events [49]. Optimization based approaches have tried to cast episodic adaptation as an explicit optimization procedure either by treating the optimizer as a black-box system [59, 60] or by learning a synaptic configuration such that one or a few gradient steps are sufficient to adapt the input/output mapping to a specific task [61].

$\diamond$ **Artificial fast plasticity:** Networks with dynamic weights that can adapt as a function of neural activation have shown promising results over regular recurrent neural networks to handle sequential data [62, 29, 31, 63, 32, 34]. However, contrary to our work, these models postulate a persistent neural activity orchestrating weights evolution. On the contrary, we show that synaptic states are the sole persistent components needed to perform fast adaptation. Additionally, the possibility of optimizing synaptic dynamics with evolutionary strategies in randomly initialized networks [64] or through gradient descent [65] has been demonstrated, as well as in a time-continuous setting

[66]. Recent results have shown that plasticity rules differentially tuned at the synapse level allow to dynamically edit and query networks memory [31, 67]. However another specificity of this work is that our model synaptic rule is a function of reward and synaptic state, allowing to drive weight dynamics conditionally on both an external feedback signal and the current model belief.

⋄ **Associative memory:** As discussed above, efficient memory storage and manipulation is a crucial feature for building rapidly learning agents. To improve over vanilla recurrent neural network policies [48], some models have augmented recurrent agents with content-addressable dictionaries able to reinstate previously encoded patterns given the current state [68–70, 13]. However these slot-based memory systems are subject to interference with incoming inputs and their memory cost grows linearly with experience. Contrastingly, attractor networks can be learnt to produce fast compression of sensory information into a fixed size tensorial representation [71, 72]. One class of such network are Hopfield networks [73–76] which benefit from a large storage capacity [76], can possibly perform hetero-associative concept binding [77, 34] and produce fast and flexible information retrieval [47].

## 5  Experiments

In this section, we explore the potential of our meta-learnt synaptic update rule with respect to the three properties of the meta-RL problem exposed in section 2. Namely, 1) *efficiency*, 2) *capacity* and 3) *generality* of the produced learning algorithm. We compare it with three state-of-the-art meta-RL models based on different adaptation mechanisms: RL$^2$ [48] a memory-based algorithm based on training a GRU-cell to perform Reinforcement Learning within the hidden space of its recurrent unit, MAML [61] which performs online gradient descent on the weights of three-layer MLP and PEARL [78], which performs probabilistic task inference for conditioning policy. Details for each experimental setting are further discussed in S.I. and the code can be found at `https://github.com/mathieuchal/metods2022`.

### 5.1  Efficiency: One-shot reinforcement learning and rapid motor control

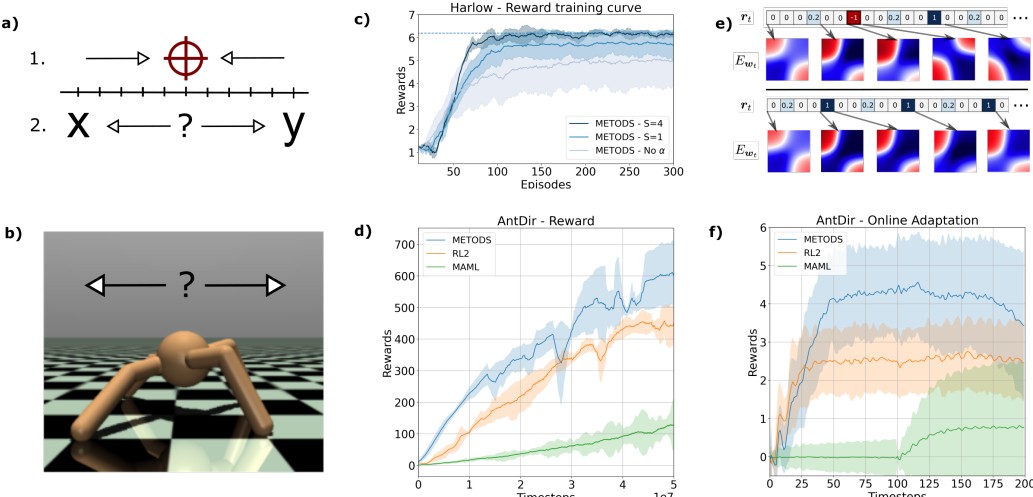

**Figure 3: a-b)** Schemas of the Harlow and Mujoco *Ant-directional* locomotion task. **c-d)** Evolution of accumulated reward over training. In the Harlow task, we conduct an ablation study by either reducing the number of recursive iterations (S=1) or removing the trainable plasticity weights $\boldsymbol{\alpha}$ resulting in sub-optimal policy. In *Ant-dir* we compare our agent training profile against MAML and RL$^2$. **e)** We can interpret the learned policy in terms of a Hopfield energy adapting with experience. We show horizontally two reward profiles of different episodes and the energy $E_{\boldsymbol{W}_t}(v_1, v_2) = -v_1^T \boldsymbol{W}_t v_2$ along two principal components of the vector trajectory $\boldsymbol{v_t}$. In the first episode, the error in the first presentation (red square) transforms the energy landscape which changes the agent policy, while on the other episode, the model belief does not change over time. Note the two modes for every energy map, which allows the model to handle the potential position permutation of the presented values. **f)** Average rewards per timestep during a single episode of the *Ant-dir* task.

To first illustrate that learnt synaptic dynamics can support fast behavioral adaptation, we use a classic experiment from the neuroscience literature originally presented by Harlow [79] and recently reintroduced in artificial meta-RL in [54] as well as a heavily-benchmarked MuJoCo directional locomotion task (see Fig. 3). To behave optimally in both settings, the agent must quickly identify the task rule and implement a relevant policy : The Harlow task consists of five sequential presentations of two random variables placed on a one-dimensional line with random permutation of their positions that an agent must select by reaching the corresponding position. One value is associated with a positive reward and the other with a negative reward. The five trials are presented in alternance with periods of fixation where the agent should return to a neutral position between items. In the MuJoCo robotic *Ant-dir* experiment, a 4-legged agent must produce a locomotive policy given a random rewarded direction.

◇ **Harlow**: Since values location are randomly permuted across presentations within one episode, the learner cannot develop a mechanistic strategy to reach high rewards based on initial position. Instead, to reach the maximal expected reward over the episode, the agent needs to perform one-shot learning of the task-contingent association rule during the first presentation. We found that even a very small network of N=20 neurons proved to be sufficient to solve the task perfectly. We investigated the synaptic mechanism encoding the agent policy. A principal component analysis reveals a policy differentiation with respect to the initial value choice outcome supported by synaptic specialization in only a few time-steps (see Figure 1-c). We can further interpret this adaptation in terms of sharp modifications of the Hopfield energy of the dynamic weights (Figure 3-e). Finally, the largest synaptic variations measured by the sum of absolute synaptic variations occur for states that carry a non-null reward signal (see S.I). These results suggest that the recursive hebbian update combined with reward feedback is sufficient to support one-shot reinforcement learning of the task association rule.

◇ **MuJoCo Ant-dir**: We trained models for performing the task on an episode of 200 timesteps and found that MetODS can adapt in a few time-steps, similar to memory-based models such as $RL^2$, (Figure 3-f) thanks to its continual adaptation mechanism. By design, MAML and PEARL do not present such a property, and they need multiple episodes before being able to perform adaptation correctly. We still report MAML performance after running its gradient adaptation at timestep t=100. We further note that our agent overall performance in a single episode is still superior to MAML performance reported in [61, 49] when more episodes are accessible for training.

## 5.2 Capacity : Maze exploration task

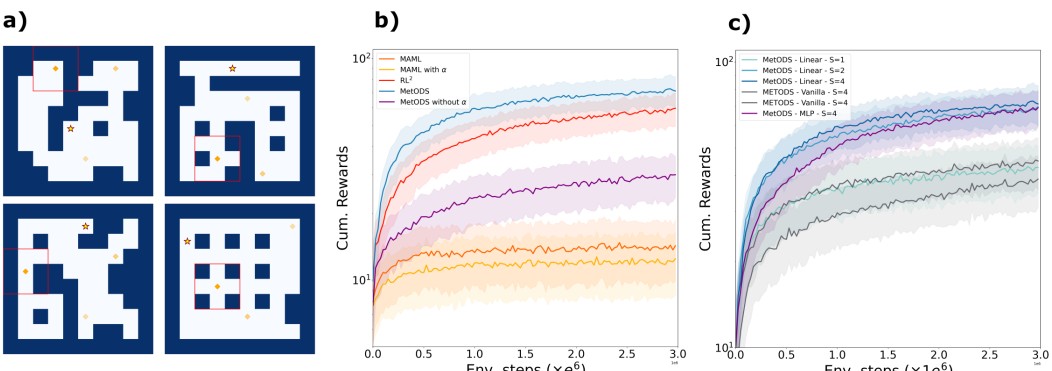

**Figure 4: a)** Examples of generated maze configurations. The mazes consist in 8 × 8 pix. areas, with walls and obstacles randomly placed according to a variation of Prim's algorithm [80] with target location randomly selected for the entire duration of the episode (star). The agent receptive field is highlighted in red. **b)** Comparisons with MAML and RL$^2$ and effect of element-wise plasticity $\alpha$. **c)** Variations of MetODS writing mechanism as well as depth S of recursivity yield different performances (see. S.I for further details)

We further tested the systematicity of MetODS learnt reinforcement program on a partially observable Markov decision process (POMDP): An agent must locate a target in a randomly generated maze while starting from random locations and observing a small portion of its environment (depicted in Figure 4). While visual navigation has been previously explored in meta-RL [48, 49, 31], we here focus on the mnemonic component of navigation by complexifying the task in two ways, we reduce the agent's visual field to a small size of 3x3 pix. and randomize the agent's position after every reward encounter. The agent can take discrete actions in the set {up,down,left,right} which moves

it accordingly by one coordinate. The agent's reward signal is solely received by hitting the target location, thus receiving a reward of 10. Each time the agent hits the target, its position is randomly reassigned on the map (orange) and the exploration resumes until 100 steps are accumulated during which the agent must collect as much reward as possible. Note that the reward is invisible to the agent, and thus the agent only knows it has hit the reward location because of the activation of the reward input. The reduced observability of the environment and the sparsity of the reward signal (most of the state transitions yield no reward) requires the agent to perform logical binding between distant temporal events to navigate the maze. Again, this setting rules out PEARL since its latent context encoding mechanism erases crucial temporal dependencies between state transitions for efficient exploration. Despite having no particular inductive bias for efficient spatial exploration or path memorization, a strong policy emerges spontaneously from training.

| Agent | 1st rew.* ($\downarrow$) | Success ($\uparrow$) | Cum. Rew. ($\uparrow$) | Cum. Rew (Larger) ($\uparrow$) |
|---|---|---|---|---|
| Random | $96.8 \pm 0.5$ | $5\%$ | $3.8 \pm 8.9$ | $3.7 \pm 6.4$ |
| MAML | $64.3 \pm 39.3$ | $45.2\%$ | $14.95 \pm 4.5$ | $5.8 \pm 10.3$ |
| RL$^2$ | $16.2 \pm 1.1$ | $96.2\%$ | $77.7 \pm 46.5$ | $28.1 \pm 29.7$ |
| MetODS | $\mathbf{14.7 \pm 1.4}$ | $\mathbf{96.6\%}$ | $\mathbf{86.5 \pm 46.8}$ | $\mathbf{34.9 \pm 34.9}$ |

**Figure 5:** Performance of Meta-RL models tested at convergence ($1e^7$ env. steps). MetODS better explores the maze as measured by the average number of steps before 1st reward and the success rate in finding the reward at least once. It then better exploits the maze as per the accumulated reward. (* We assign 100 to episodes with no reward encounter.)

$\diamond$ **Ablation study and variations:** We explored the contribution of the different features combined in MetODS update rule, showing that they all contribute to the final performance of our meta-learning model. First, we tested the importance of the element-wise tuning of plasticity in weight-based learning models and note that while it adversely affects MAML gradient update, it greatly improves MetODS performance, suggesting different forms of weights update. Second, we verified that augmenting recursivity depth S was beneficial to performance, consistently with Harlow results. Third, we noted that transforming the rightmost vector of the writing equation in 5 with a linear projection (*Linear* in figure 4, see S.I for full experimental details) yields major improvement while non-linear (*MLP*) does not improve performance. Finally, we additionally test the capability of the learnt navigation skills to generalize to a larger maze size of $10 \times 10$ pix. unseen during training. We show that MetODS is able to retain its advantage (see figure 4 and table 5 for results).

## 5.3 Generality : Motor control

Finally, we test the generality of the reinforcement learning program learnt by our model for different continuous control tasks:

**MetaWorld:** First, we use the dexterous manipulation benchmark proposed in [81] using the benchmark suite [82], in which a Sawyer robot is tasked with diverse operations. A full adaptation episode consists in N=10 rollouts of 500 timesteps of the same task across which dynamic weights are carried over. Observation consists in the robot's joint angles and velocities, and the actions are its joint torques. We compare MetODS with baseline methods in terms of meta-training and meta-testing success rate for 3 settings, *push, reach* and *ML-10*. We show in Fig. 6 the meta-training results for all the methods in the MetaWorld environments. Due to computational resource constraints, we restrict our experiment to a budget of 10M steps per run. While we note that our benchmark does not reflect the final performance of previous methods reported in [81] at 300M steps, we note that MetODS test performance outperforms these methods early in training and keeps improving at 10M steps, potentially leaving room for improvement with additional training. (see S.I for additional discussion). Finally, we note that all tested approaches performed modestly on ML10 for test tasks, which highlights the limitation of current methods. We conjecture that this might be due to the absence of inductive bias for sharing knowledge between tasks or fostering systematic exploration of the environment of the tested meta-learning algorithms.

**Robot impairment:** We also tested the robustness of MetODS learnt reinforcement programs by evaluating the agent ability to perform in a setting not seen during training: specifically, when partially impairing the agent motor capabilities. We adopt the same experimental setting as section 5.1 for the *Ant* and *Cheetah* robots and evaluate the performance when freezing one of the robots

torque. We show that our model policy retains a better proportion of its performance compared to other approaches. These results suggest that fast synaptic dynamics are not only better suited to support fast adaptation of a motor policy in the continuous domain, but also implement a more robust reinforcement learning program when impairing the agent motor capabilities.

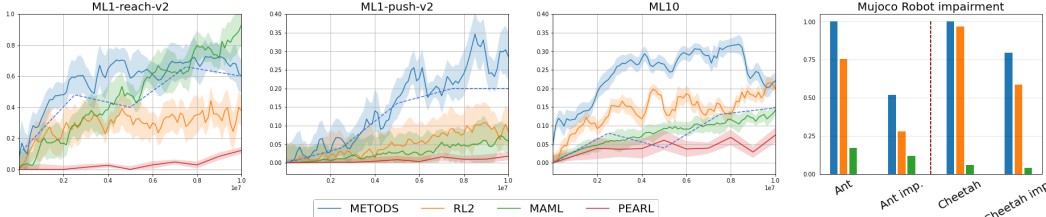

**Figure 6: Left** Meta-training results for MetaWorld benchmarks. Subplots show tasks success rate over training timesteps. Average meta-test results for MetODS is shown in dotted line. **Right** Cumulative reward of the *Ant* and *Cheetah* directional locomotion task. For each condition, results are normalized against the best performing policy

## 6 Discussion

In this work, we introduce a novel meta-RL system, MetODS, which leverages a self-referential weight update mechanism for rapid specialization at the episodic level. Our approach is generic and supports discrete and continuous domains, giving rise to a promising repertoire of skills such as one-shot adaptation, spatial navigation or motor coordination. MetODS demonstrates that locally tuned synaptic updates whose form depends directly on the network configuration can be meta-learnt for solving complex reinforcement learning tasks. We conjecture that further tuning the hyperparameters as well as combining MetODS with more sophisticated reinforcement learning techniques can boost its performance. Generally, the success of the approach provides evidence for the benefits of fast plasticity in artificial neural networks, and the exploration of self-referential networks.

## 7 Broader Impact

Our work explores the emergence of reinforcement-learning programs through fast synaptic plasticity. The proposed pieces of evidence that information memorization and manipulation as well as behavioral specialization can be supported by such computational principles 1) help question the functional role of such mechanisms observed in biology and 2) reaffirms that alternative paradigms to gradient descent might exists for efficient artificial neural network control. Additionally, the proposed method is of interest for interactive machine learning systems that operates in quickly changing environments and under uncertainty (bayesian optimization, active learning and control theory). For instance, the meta RL approach proposed in this work could be applied to brain-computer interfaces for tuning controllers to rapidly drifting neural signals. Improving medical applications and robot control promises positive impact, however deeper theoretical understanding and careful deployment monitoring are required to avoid misuse.

## 8 Acknowledgments and Disclosure of Funding

This work was supported by ANR-3IA Artificial and Natural Intelligence Toulouse Institute (ANR-19-PI3A-0004), OSCI-DEEP ANR (ANR-19-NEUC-0004), ONR (N00014-19-1-2029), and NSF (IIS-1912280 and EAR-1925481). Additional support provided by the Carney Institute for Brain Science and the Center for Computation and Visualization (CCV) via NIH Office of the Director grant S10OD025181. The authors would like to thank the anonymous reviewers for their thorough comments and suggestions that led to an improved version of this work.

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
