# Supplementary Information : Meta-Reinforcement Learning with Self-Modifying Networks

## 9 Optimization

Defining the weight parameters $\boldsymbol{W}$ of MetODS as dynamic variables lifts the optimization problem (2) into a functional space of control functions parameterized by $\boldsymbol{\theta}$. Hence, meta-optimizing the control necessitates the estimation of gradients with respect to $\boldsymbol{\theta}$ over the space $\mathbb{T}$ and for any possible trajectory $\pi_t$ in $\Pi$. Interestingly, previous meta RL approaches have performed policy gradient optimization by sampling a single policy trajectory $\pi_t \sim \mathcal{M}_{\boldsymbol{\theta}}(\tau)$ over $M$ multiple tasks, showing that it is sufficient to obtain correct gradient estimates on $\boldsymbol{\theta}$. We proceed in the same way, by estimating the gradient policy update integrated over the space of tasks as mini-batches over tasks.

$$\frac{\partial}{\partial \boldsymbol{\theta}} \mathbb{E}_{\tau \sim \mu_{\mathbb{T}}} \left[ \mathbb{E}_{\pi \sim \mu_{\pi}^{\boldsymbol{\theta},\tau,t}} \left[ \mathcal{R}(\tau, \pi) \right] \right] \quad \approx \quad \sum_{\tau_1,\ldots,\tau_n} \sum_{t=0}^{T} \frac{\partial \log \pi_t(\boldsymbol{a}_t | \boldsymbol{W}_t, \boldsymbol{\theta})}{\partial \boldsymbol{\theta}} r_{\tau_i}(\boldsymbol{a}_t, \boldsymbol{s}_t) \tag{7}$$

Additionally, the memory cost of storing synaptic weights trajectories instead of hidden activity in a network of $N$ neurons is $\mathcal{O}(N^2)$ instead of $\mathcal{O}(N)$. This might lead to prohibitively large memory requirements for training with BPTT [83] over long episodes. We present in S.I an alternative solution to train the model through the discrete adjoint sensitivity method, leveraging the work of [84] yielding a memory cost of $\mathcal{O}(1)$. The agent's log-policy total derivative with respect to $\boldsymbol{\theta}$ can be computed as the solution of an augmented adjoint problem [67].

### 9.1 Gradient policy update

We define the evolution of the agent policy in task $\tau$ up to step $T$ as the stochastic policy process $(\pi_t)_{t \leq T}$ in the space of policy $\Pi$ with measure $\mu_{\pi}^{\boldsymbol{\theta},\tau,1\cdots T}$ and write $(\boldsymbol{W}_t)_{t \leq T}$ the trajectory of weights, such that:

$$\left( \pi_t \right)_{t \leq T} = \left( \pi_t(\cdot | \boldsymbol{W}_t, \boldsymbol{s}_t, \boldsymbol{\theta}) \right)_{t \leq T} \sim \mu_{\pi}^{\boldsymbol{\theta},\tau,1\cdots T} \tag{8}$$

We also recall the definition of $\mathcal{R}(\tau, \pi)$ as the average accumulated reward under the realisation of $(\pi_t)_{t \leq T}$ for task $\tau$. Then, The policy gradient update used to train MetODS can be written as the average gradient of $\boldsymbol{\theta}$ with respect to $\mu_{\mathbb{T}}$ and $\mu_{\pi}^{\boldsymbol{\theta},\tau,1\cdots T}$:

$$\frac{\partial}{\partial \boldsymbol{\theta}} \, \mathbb{E}_{\tau \sim \mu_{\mathbb{T}}} \left[ \mathbb{E}_{\pi \sim \mu_{\pi}^{\boldsymbol{\theta},\tau,1\cdots T}} \left[ \mathcal{R}(\tau, \pi) \right] \right] = \mathbb{E}_{\tau \sim \mu_{\mathbb{T}}} \left[ \sum_{t=0}^{T} \frac{\partial}{\partial \boldsymbol{\theta}} \mathbb{E}_{\pi_t, \mathcal{P}_\tau} \left[ r_\tau(\boldsymbol{a}_t, \boldsymbol{s}_t) \right] \right] \tag{9}$$

Then for any $t \leq T$, we can rewrite the gradient of the average, using the log-policy trick:

$$\frac{\partial}{\partial \boldsymbol{\theta}} \mathbb{E}_{\pi_t, \mathcal{P}_\tau} \left[ r_\tau(\boldsymbol{a}_t, \boldsymbol{s}_t) \right] = \mathbb{E}_{\pi_t, \mathcal{P}_\tau} \left[ r_\tau(\boldsymbol{a}_t, \boldsymbol{s}_t) \frac{\partial \log \pi_t(\boldsymbol{a}_t^\tau | \boldsymbol{W}_t, \boldsymbol{\theta})}{\partial \boldsymbol{\theta}} \right] \tag{10}$$

As specified in section 3.3, we do not sample over the policy distribution $\pi_t$ and probability transition $\mathcal{P}_\tau$ to estimate the inner expectation in (11). Instead, we rely on a single evaluation, which yield, combining (9) with (10):

$$\frac{\partial}{\partial \boldsymbol{\theta}} \, \mathbb{E}_{\tau \sim \mu_{\mathbb{T}}} \left[ \mathbb{E}_{\pi \sim \mu_{\pi}^{\boldsymbol{\theta},\tau,1\cdots T}} \left[ \mathcal{R}(\tau, \pi) \right] \right] \quad \approx \quad \sum_{t=0}^{T} \left[ r_\tau(\boldsymbol{a}_t, \boldsymbol{s}_t) \frac{\partial \log \pi_t(\boldsymbol{a}_t^\tau | \boldsymbol{W}_t, \boldsymbol{\theta})}{\partial \boldsymbol{\theta}} \right] \tag{11}$$

By sampling over tasks, this last equation allows us to write the following gradient estimator

$$\frac{\partial}{\partial \boldsymbol{\theta}} \mathbb{E}_{\tau \sim \mu_{\mathbb{T}}} \left[ \mathbb{E}_{\pi \sim \mu_{\pi}^{\boldsymbol{\theta},\tau,1\cdots T}} \left[ \mathcal{R}(\tau, \pi) \right] \right] \quad \approx \quad \frac{1}{M} \sum_{i=0}^{M} \sum_{t=0}^{T} \frac{\partial \log \pi_t(\boldsymbol{a}_t^{\tau_i} | \boldsymbol{W}_t, \boldsymbol{\theta})}{\partial \boldsymbol{\theta}} r(\boldsymbol{a}_t^{\tau_i}, \boldsymbol{s}_t^{\tau_i}) \tag{12}$$

## 9.2 Discrete adjoint system

With the previous notation, let us define the total gradient function $\frac{\partial \mathcal{J}}{\partial \boldsymbol{\theta}}$ as the gradient of our objective function:

$$\frac{\partial \mathcal{J}}{\partial \boldsymbol{\theta}}(\boldsymbol{\theta}) = \frac{1}{M} \sum_{t=0}^{T} \sum_{i=0}^{M} \frac{\partial \log \pi_t(\boldsymbol{a}_t^{\tau_i}|\boldsymbol{W}_t, \boldsymbol{\theta})}{\partial \boldsymbol{\theta}} r(\boldsymbol{a}_t^{\tau_i}, \boldsymbol{s}_t^{\tau_i}) \tag{13}$$

To clarify, we shall introduce the intermediary cost notation:

$$c_t(\boldsymbol{W}, \boldsymbol{\theta}, t) = \sum_{i=0}^{M} \log \pi_t(\boldsymbol{a}_t^{\tau_i}|\boldsymbol{W}_t, \boldsymbol{\theta}) r(\boldsymbol{a}_t^{\tau_i}, \boldsymbol{s}_t^{\tau_i}) \tag{14}$$

and we note the following update equation

$$\boldsymbol{W}_{t+1} = \delta(\boldsymbol{W}_t, \boldsymbol{\theta}) \tag{15}$$

This identify a discrete dynamical system with finite sum and differentiable cost, whose gradient can be computed mediating the introduction of an *adjoint* dynamical system presented in section 2 of [84] . Defining $(\boldsymbol{\nu}_t)_{t \leq T}$ the adjoint sequence, the general gradient equation can be computed as:

$$\frac{\partial \mathcal{J}}{\partial \boldsymbol{\theta}}(\boldsymbol{\theta}) = \left[\frac{\partial c_0}{\partial \boldsymbol{W}_0} - \boldsymbol{\nu}_0 \cdot \frac{\partial \delta}{\partial \boldsymbol{W}_0} - \boldsymbol{\nu}_0\right]^{\dagger} \frac{\partial \boldsymbol{W}_0}{\partial \boldsymbol{\theta}} + \sum_{i=1}^{T} \left[\frac{\partial c_t}{\partial \boldsymbol{\theta}} - \boldsymbol{\nu}_t^{\dagger} \cdot \frac{\partial \delta}{\partial \boldsymbol{\theta}}\right] \tag{16}$$

Applying formula (16) to MetODS yields the following gradient formula:

$$\frac{\partial \mathcal{J}}{\partial \boldsymbol{\theta}}(\boldsymbol{\theta}) = \left[\frac{\partial c_0}{\partial \boldsymbol{W}_0} - \boldsymbol{\nu}_0\right] + \sum_{i=1}^{T} \left[\frac{\partial c_t}{\partial \boldsymbol{\theta}} - \boldsymbol{\nu}_t^{\dagger} \cdot \frac{\partial \delta}{\partial \boldsymbol{\theta}}\right] \tag{17}$$

where $(\boldsymbol{\nu}_t)_{t \leq T}$ follows the following update rule backwards:

$$\begin{cases} \boldsymbol{\nu}_T = 0 \\ \boldsymbol{\nu}_{t-1} = \boldsymbol{\nu}_t - \frac{\partial c_t}{\partial \boldsymbol{W}_t} \end{cases} \tag{18}$$

# 10  Experiment details

**General information:**  As specified in section **3** and **5**, we test a single model definition for all experiments in this work, with one layer of dynamic weights $\boldsymbol{W}_t$. This layer consists in a dense matrix of size $\boldsymbol{n} \in \mathbb{N}$ with learnt or random initialization. Our model lightweight parametrization of the synaptic update rule makes it a very parameter efficient technique, which can perform batch-computation and be ported to GPU hardware to accelerate training. We refer readers to table 1 for specific details of each experiment presented in section 5. In addition to this *pdf* file, we provide the code for training METODS agents in experiments presented in section 5 at https:/

| FEATURE - EXPERIMENT | HARLOW | GYM MUJOCO | MAZE | META-WORLD |
|---|---|---|---|---|
| DYNAMIC LAYER SIZE $n$ | 20 | 100 | 200 | 100 |
| INPUT SIZE $i$ | 12 | 134 (ANT) / 27 (CHEETAH) | 15 | I |
| OUTPUT SIZE $o$ | 2 | 8 (ANT)/ 6 (CHEETAH) | 4 | 8 |
| EMBEDDING $f$ | $[i \times 32, \sigma, 32 \times n]$ | $[i \times 64, \sigma, 64 \times n]$ | $[i \times 32, \sigma, 32 \times n]$ | $[i \times 32, \sigma, 32 \times 64, \sigma, 64 \times n]$ |
| READ-OUT $g$ | $[n \times 32, \sigma, 32 \times o]$ | $[n \times 64, \sigma, 64 \times o]$ | $[n \times 32, \sigma, 32 \times o]$ | $[n \times 64, \sigma, 64 \times o]$ |
| NON-LINEARITIES | TANH | TANH | TANH | TANH |
| INIT. $\boldsymbol{W}_0$ | $\mathcal{N}(0, 1e-3)$ | LEARNED | LEARNED | LEARNED |
| # NUM. OF EPISODES | 5 (MAX) | 1 | 1 | 10 |
| LENGHT OF 1 EPISODE | 250 (MAX) | 200 | 100 | 500 |
| META-TRAINING ALG. | A2C | A2C | A2C | PPO |
| LEARNING RATE | 5E-4 | 1E-4 | 5E-4 | 5E-4 |
| META-BATCH-SIZE | 50 | 50 | 20 | 25 |
| DISCOUNT FACTOR $\lambda$ | 9E-1 | 9E-1 | 9.9E-1 | 9.9E-1 |
| GAE | 1. | 9.5E-1 | 9.5E-1 | 9.5E-1 |
| VALUE FUNCTION COEFF. | 4E-1 | 4E-1 | 4E-1 | - |
| ENTROPY REG. FACTOR | 3E-2 | 3E-2 | 1E-2 | 1E-2 |

**Table 1:** Summary of training hyper-parameters for the four experiments presented in this work.

**Meta-training algorithm:** We show in our experiments, that the all meta-parameters $\boldsymbol{\theta} = [\boldsymbol{\alpha}, \boldsymbol{\kappa}, \boldsymbol{\beta}, \boldsymbol{f}, \boldsymbol{g},]$ can be jointly optimized with two policy gradient algortihms. We use policy gradient methods to meta-train the synaptic parameters: Advantage Actor-critic algorithm (A2C) [85] where we consider that temporal dependancies are crucial to solve the task, and we use PPO [86] as a sequential policy optimization over fixed rollouts for the motor control experiment. Additionally, to reduce noise in policy gradient updates, we further show that it is possible to learn a dynamic advantage estimate of the Generalized Advantage Estimation (GAE) [87] and that we can meta-learn it as a second head of the MetODS layer output $\boldsymbol{v}^{(s)}$. This fact confirms that tuned hebbian-updates are also a sufficient mechanism to keep track of a policy value estimate.

**Plasticity parameters:** In all our experiments, $\boldsymbol{\alpha}$ consists in a real-valued matrix of $\mathbb{R}^{n \times n}$ initialized with independent normal distribution $\mathcal{N}(\mu = 0, \sigma = 1e-3)$. Similarly, multi-step weighting parameters $(\boldsymbol{\kappa}_s^{(l)})_{l \leq s}$ and $(\boldsymbol{\beta}_s^{(l)})_{l \leq s}$ can be stored as entries of triangular inferior matrices of $\mathbb{R}^{S \times S}$ and are initialized with $\mathcal{N}(\mu = 0, \sigma = 1e-2)$.

**Embedding and read-out:** At each time-steps, inputs to the feed-forward embedding function $\boldsymbol{f}$ consist in a concatenation of new observable/state $\boldsymbol{s}_t$ as well as previous action and reward $\boldsymbol{a}_{t-1}$ and $\boldsymbol{r}_{t-1}$. After adaptation procedure, policy is read-out from the last activation vector $\boldsymbol{v}^{(S)}$ by a feed-forward function $\boldsymbol{g}$ which output statistics of a parameterized distribution in the action space (categorical with Softmax normalization in the discrete case or Gaussian in the continuous case). Both input and output mappings $\boldsymbol{f}$ and $\boldsymbol{g}$ consist in 2-layer Perceptrons with dense connections initialized with orthogonal initialization and hyperbolic tangent non-linearities.

## 10.1 Harlow task

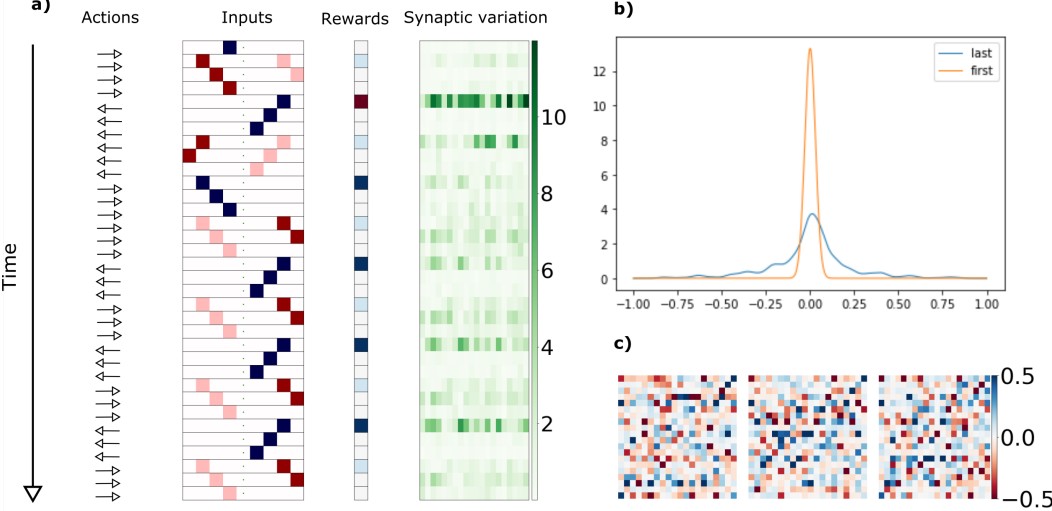

**Figure 7: Harlow task: a)** An episode of the Harlow task. From left to right: Actions, visual field of the agent (blue squares:fixation target, red squares: values), Rewards (red;negative,light blue: fixation reward, dark blue:value reward) and sum of absolute synaptic variation $\| \sum \Delta \boldsymbol{W} \|$ per neuron. **b)** Distribution of weights before and after adaptation over 5 presentations.**c)** Three instances of the dynamic weights $\boldsymbol{W}_t$ after adaptation that solved the Harlow task. Every synaptic configuration presents differences but perform optimally.

This experiment consists in a 1-dimensional simplification of the task presented in [54] and inspired from https://github.com/bkhmsi/Meta-RL-Harlow. The action space is the discrete set $\{-1, 1\}$ moving the agent respectively to the left/right on a discretized line. The state space consist in 17 positions while the agent receptive field is eight dimensional. Values are placed at 3 positions from the fixation target position. The fixation position yields a reward of 0.2 while the values are drawn uniformly from $[\![0, 10]\!]$ and randomly associated with a reward of $-1$ and 1 at the beginning of each episode. The maximal duration for an episode is 250 steps, although the model solves the 5 trials in $\sim 35$ steps on average.

## 10.2 Gym Mujoco directional robot control

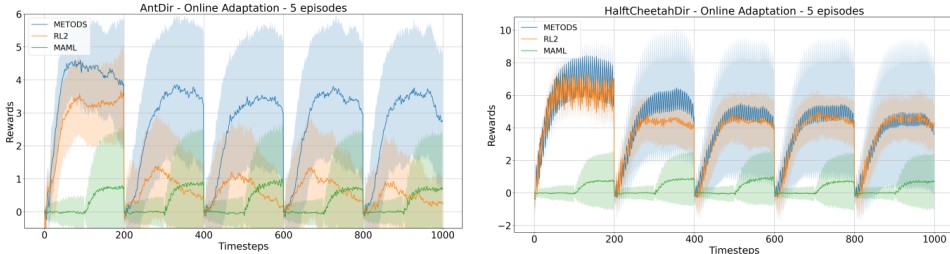

**Figure 8:** Reward profiles of MAML, RL$^2$ and MetODS over 5 consecutive episodes with randomly varying rewarded direction. In *Cheetah*, MetODS slightly overperforms RL$^2$ and in *Ant*, MetODS is the only approach able to reach performance comparable to first episode in subsequent episodes.

**Setting:** We consider the directional rewards task proposed in [61] with the *Ant* and *Cheetah* robots, as a more complex test of rapid adaptation. We apply standard RL practice for continuous control by parameterizing the stochastic policy as a product of independent normal distributions with fixed standard deviation $\sigma = 0.1$ and mean inferred by the agent network. A training meta-episode consists in a single rollout of 200 steps with random sampling of the reward direction for each episode.

**Robot impairment:** We partially impaired the agent motor capabilities by "freezing" one of the robot actuator. Namely we consistently passed a value of zero to the the *right_back_leg* in Ant (coordinate 8 in OpenAI Gym XML asset files) and *ffoot* in Cheetah (coordinate 6 in OpenAI Gym XML asset files).

**Continual adaptation:** Finally, we further investigated whether the adaptation mechanism was also adjustable after performing initial adaptation by testing trained agents over 5 consecutive episodes with randomly changing rewarded direction while retaining weight state across episodes. We show in Fig 10 the average reward profile of the three meta-RL agents over 1000 test episodes, where MetODS adapt faster and better with respect to reward variations.

## 10.3 Maze Navigation task

**Maze generation:** The maze environments are created following Prim's algorithm [80] which randomly propagates walls on a board of $N \times N$ cells. We additionally add walls to cell location were no walls has been created at any of the neighboring 8 cells to avoid null inputs to the agent. The reward and agent locations are selected at random at the beginning of an episode. The reward location does not change during the episode while the agent is restarting from a uniform random location after every encounter with reward.

**Models capacity:** In this experiment, since model memory capacity seems to directly impact final performance, we specifically control for the number of learnable parameters in order to compare models and fix a training budget of 10M env. steps. Since MAML does not have a continual adaptation mechanism, we perform gradient adaptation every 20 time-steps during

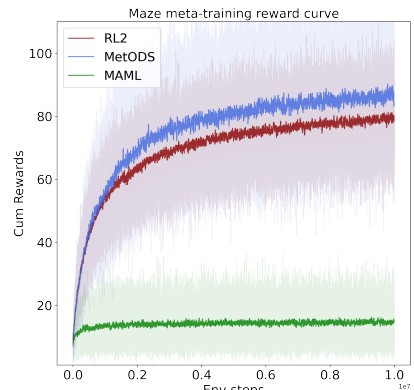

**Figure 9:** Meta-training rewards curves of the maze experiment presented in section 5.2

an episode, in order to balance noisiness of the gradient and rapidity of exploration. We show in Table 2 the mean over 1000 episodes of the accumulated reward of agents trained on $8 \times 8$ mazes and tested on different maze sizes $N \in (4, 6, 8, 10, 12)$. Agents performances are highly varying within a size setting due to differences in maze configurations and across maze sizes due to increasing complexity. However, MetODS and RL$^2$ are able to generalize at least partially to these new settings (see table 2). MetODS outperforms RL$^2$ in every setting, and generalizes better, retaining an advantage of up to $25\%$ in accumulated reward in the biggest maze.

| MAZE SIZE | MAML | RL$^2$ | METODS | REL IMP. TO RL$^2$ |
|---|---|---|---|---|
| 6 | $21.0 \pm 18.4$ | $149.3 \pm 66.7$ | $\mathbf{169.1 \pm 66.1}$ | 13% |
| 8 | $14.9 \pm 4.5$ | $72.1 \pm 45.6$ | $\mathbf{87.3 \pm 48.3}$ | 20% |
| 10 | $5.7 \pm 7.9$ | $28.1 \pm 29.7$ | $\mathbf{34.9 \pm 34.9}$ | 21% |
| 12 | $3.9 \pm 6.9$ | $11.1 \pm 15.8$ | $\mathbf{13.9 \pm 19.8}$ | 25% |

**Table 2:** Average accumulated reward over 1000 episodes for different maze sizes for agents trained for N=8.

Additionally, we investigated the selectivity of neurons of MetODS plastic weight layer with respect to spatial location, to investigate whether there could be emergence of activation patterns resembling those of grid cells found in the mammalian entorhinal cortex. We measured the selectivity of each neuron for specific agent locations by measuring the average activation rate normalized by the number of agent passages in different maze configurations. Interestingly, we found that without any explicit regularization, some neurons displayed sparse activation and consistent remapping between maps.

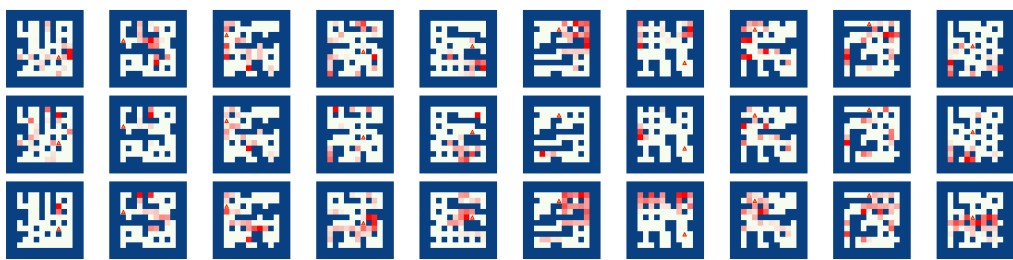

**Figure 10:** Heatmap of activation rate of three neurons of a trained model across 10 different maze maps normalized by agent passage. Activation patterns show selectivity to particular positions within an episode and strong remapping across maze configurations.

## 10.4 Meta-world

**Experiments:** In order to run experiments and baseline of MetaWorld [81], we used the training routines and functions offered in the python library GARAGE [82]. We ran two different types of experiments:

1. **ML1**: In this setting, we restrict the task distribution to a single type of robotic manipulation while varying the goal location. The meta-training "tasks" corresponds to 50 random initial agent and goal positions, and meta-testing to 50 heldout positions. We tested our model on the *reach-v2* and *push-v2* tasks.

2. **ML10**: This set-up tests generalization to new manipulation tasks, the benchmark provides 10 training tasks and holds out 5 meta-testing tasks.

**Training details:** We used PPO [86] for training our model, with clip range $r = 0.2$ and 10 inner optimization steps maximum, for its good empirical performance and in order to compare with other methods with recurrent computation scheme. We kept the experimental settings from the original paper with N=10 episodes of 500 time-steps and sampled proximal gradient update in a meta-batch size of 25 trajectories. To estimate value function, we both tested a feedforward neural network trained at each iteration of the meta-training or an additional MetODS network, which resulted in similar training results.