# OpenReview forum: "Meta-Reinforcement Learning with Self-Modifying Networks"
_NeurIPS.cc/2022/Conference — NeurIPS 2022 Accept_

### Official Review · Reviewer_7zkD · 2022-07-02

**Rating:** 6
**Confidence:** 2
**Soundness:** 2 fair
**Presentation:** 2 fair
**Contribution:** 3 good

**Summary:**

This paper presents a new neural network that is able to modify its own weights, known as the self-modifying network.
By incorporating self-modifying networks into the meta reinforcement learning (RL) framework, the new model Meta-Optimized Dynamical Synapses (MetODS) shows better empirical performance in terms of efficiency, capacity, and generality, compared to several meta RL baseline algorithms.

**Questions:**

What neural network structures are used for baseline algorithms such as MAML and RL2?

**Limitations:**

I may misunderstand something here, but in equation 5, it seems that computing $v^{(s)}$ and $W^{(s)}$ requires whole trajectories of $v^{(l)}$ and $W^{(l)}$ to be stored, which leads to a large memory requirement.
The discrete adjoint sensitivity method only helps with the backward pass but not the forward pass (i.e. computing $v^{(s)}$ and $W^{(s)}$). How about the forward pass?

**Strengths And Weaknesses:**

This work is well motivated. The proposed algorithm MetODS shows significantly better performance across different kinds of tasks, from maze navigation to motor control.

Nevertheless, It is hard for me to understand the reasons behind the success of self-modifying networks and MetODS.
In Section 3, it is claimed that "Our model learns to train itself by updating its weights through interaction with the environment and its own current weight state. This mechanism enables MetODS to rapidly compress experience of a task $\tau$ into a particular synaptic configuration ......" Arguably, an RL model using traditional neural networks optimized with SGD also update its weights through interactions with the environment, based on its current weights. What is the critical difference between self-modifying networks and classical neural networks? More explanations and analytical experiments are needed to show the advantage of self-modifying networks.

Small typos:
- Line 48, conjonction --> conjunction.
- Line 11 in Algorithm 1.
- Missing one citation in Line 296.
- Missing x & y labels in Figure 6.

Finally, I do not find Figure 1 very helpful.

---

> ### Author Response · Authors · 2022-08-01
> **Reply to your review and additional clarification**
>
> **Thank you for your interest in this work as well as for your questions that notably helped us clarify our model presentation (section 3 - MetODS). Indeed, it seems that your main call, converging with other reviewers comments, is for gaining further insight into the original computational mechanisms explored by MetODS. Hence, we would like to clarify at the algorithmic level and justify at the theoretical level by:**
>  - **Answering more specifically your questions below.**
>  - **Proposing an updated version of our work where we explain more in depth the motivation and mechanisms for MetODS (notably in section 3).**
>
> [...] What is the critical difference between self-modifying networks and classical neural networks? [...]
>
> > **We agree with you on the fact that a “RL model using traditional neural networks optimized with SGD also update its weights through interactions with the environment, based on its current weights”, however, we believe that the learning rule presented in MetODS radically differs from gradient-based learning:**
>
>  > - **In SGD, the analytical expression of the update rule is “rigid” in the sense that it depends on the weight state solely through the error signal coming from a predefined loss function (More specifically, through the chain rule, it will be an affine function of the error with respect to activations $\Delta(W) = \frac{\partial a}{\partial W}.\frac{\partial \mathcal{L}}{\partial a}$). Here, we take a different approach, by considering a local update rule that depends on the current weight state through the non-linear recursive read-write scheme. This allows to make the expression of the update rule a potentially much more sensitive function of the weight content itself. This self-reflexive property of MetODS update rule is furthermore strengthened by a synaptic-wise parametrisation, which allows synapses to be differentially sensitive to this rule, which is not the case in gradient descent. We show in additional results of our revised version that this synaptic-wise parametrisation is a crucial component of our found learning rule and that is interestingly, does not benefit gradient based Meta-RL methods such as MAML. (section 5 - Maze experiments)**
>
> >  - **Moreover, while being principled in terms of convergence, SGD is not biologically plausible and separate inference from learning. Our adaptation is on the other hand, continuous with respect to the agent interactions with environment, which allows the agent to track closely state transitions structures, and articulate temporal strategies, similar to memory-based models such as RL2. This is one key feature that we believe crucial for performance in tasks were sequential planning matters such as in the maze experiment.**
>
> "What neural network structures are used for baseline algorithms such as MAML and RL2?"
>
> > **This is an important question also converging with other reviewers comments. We will dedicate a section to more clearly describe the baseline implementations in the updated submission: In all experiments, we followed original implementation: MAML is based on 3-layer deep fully-connected neural network with ReLU non linearities consistent with original implementation, while RL2 consist in the implementation of GRU cell. Particularly for the MetaWorld experiment, we use the official implementations from the _Garage_ library in order to ensure consistency with previously reported results in the literature.**
>
> Small typos:
>
> > **Thanks for raising these typos. We correct them in the updated submission version. Regarding figure 1, we will adjust it to better reflect our explanation from the first question.**
>
> "[...] I may misunderstand something here, but in equation 5, it seems that computing v(s) and W(s) requires whole trajectories of v(l) and W(l) to be stored, which leads to a large memory requirement.[...]"
>
> > **At each time-step we apply S times the recursive scheme, starting from [v^(0),W_t] and gathering intermediary versions [(v^(l),W^(l)]. After S iterations, we simply discard the intermediaries tensors as they are no longer needed and simply store W^(S) as W_{t+1}. Hence, the memory cost of such a procedure is not that important as we only have to keep at most S matrices into memory and 1) S does not need to be very large (we set S=4 for our experiments) and 2) the memory cost is not growing as a function of the episode length and 3) we are dealing with very lightweight model here as we only have a single hidden layer. Hence, our model memory requirement remain largely manageable even with mini-batching. Additionally, note that the adjoint sensitivity method has at least the same memory cost with respect to dynamic variables in backward than in forward (as it integrates the same dynamic but reversed in time) so there is no difference in this regard.**

---

> > ### Comment · Reviewer_7zkD · 2022-08-05
> > **Thank you for your clarifications**
> >
> > Thank you for your clarifications! Most of my concerns are addressed and I am willing to improve my score accordingly.
> >
> > However, I still feel that the advantage of this approach is not explained clearly enough. The reply provides some intuitive explanations which are helpful but not convinced enough. Maybe providing some small analytical experiments would be more insightful.

---

### Official Review · Reviewer_aCER · 2022-07-05

**Rating:** 6
**Confidence:** 4
**Soundness:** 2 fair
**Presentation:** 2 fair
**Contribution:** 3 good

**Summary:**

Post-response update
----------------

The authors have improved the paper and addressed some of my concerns; while I still find the assessment to be problematic in some areas, I think the present results, together with the interesting idea, make for a paper that could be a useful contribution to the field. I have updated my score accordingly.

Original review
--------------

This work presents an interesting and fairly novel perspective on meta-RL: having an agent adapt to a task by modifying its weights through an iterated Hebbian update process. The paper demonstrates that this approach outperforms vanilla baselines (RL^2, MAML, and sometimes PEARL) across a variety of settings that stress different aspects of adaptation. This is overall an intriguing approach that I hope to see published either here or in the future.


**Questions:**

See above. Briefly:
* Which aspects of each experiment might be unfair to the baselines? Would the baselines outperform the proposed algorithm if the experiments were run for longer?
* Which aspects of the approach are necessary (justified with ablation experiments)?
* Can the authors justify the policy transport perspective further, e.g. by comparing to the baselines with this perspective?



**Limitations:**

See above. Briefly:
* Unless I am misunderstanding something, the experiments are a limitation because the baselines are handicapped (e.g. by far less training).
* The conclusions we can draw from the paper are limited by the lack of ablation experiments, to identify which aspects of the approach are essential.
* The authors mention briefly that they leave extending their plasticity rule to multiple layers to future work. Of course, Hebbian rules don't tend to work well for training deep networks, and this point might deserve slightly more emphasis (e.g. a brief sentence in the discussion to highlight the challenge).

**Strengths And Weaknesses:**


Strengths:
The idea is interesting—it’s nice to see a fresh perspective on adaptation drawing on dynamical systems perspectives from neuroscience.
The breadth of demonstrations is fairly compelling; it’s great that the authors articulate different aspects of the meta-RL problem and test each of them.
The results on impared robots and with continual adaptation (in the appendix) are particularly intriguing, because they suggest an increased robustness inherent in this approach; however, this difference would be more compelling if the comparison algorithms were trained to matching train performance (see below).


Weaknesses (in order of importance):
* Some of the evaluation choices seem like they could unfairly handicap the baselines; therefore I am not sure how much to trust the overall conclusions of the paper.
  - The authors limit to 10M steps for the MetaWorld ML1 and ML10 environments; but the original MetaWorld paper achieved much higher performance with RL^2 and MAML after much longer training (300M steps). So the evaluation seems to artificially handicap the baselines, by training for much less time than the baseline approaches require. Indeed, in ML1-reach and ML10 the curves for one of the baselines appear to be passing the proposed MetODS approach just where the plot ends. Would the baselines perform better than MetODS if the experiments were run for longer?
  -The authors limit their approach to a single layer of adaptable weights—did they make a similar limitation for MAML? I wasn’t entirely clear from either the main text or the supplement. If the baselines are restricted compared to the original work this limitation would be a major caveat that should be stated much more explicitly in the text. Algorithms like MAML are generally developed and tested with deep networks; it would not be appropriate to test them with a different architecture without doing full hyperparameter tuning; and even with hyperparameter tuning the conclusion should come with caveats if the authors are artificially limiting the baseline approaches. It’s possible that the authors are not doing this; I am happy to be corrected if so, but in either case I would suggest they be more explicit about precisely what the baseline approaches were in the paper when they revise it.
  - Unless I’m misunderstanding both the above points, I find statements like “producing better agents than previous meta-RL approaches” or “compares favorably with prior meta-RL algorithms” to be somewhat misleading. Ideally the baseline experiments should be run using architectures comparable to the original papers, for as long as the original paper in at least a subset of the domains. At the very least the plots should show e.g. the max performance figures achieved in the original MetaWorld paper as a baseline comparison in each case (e.g. a star or dashed line with label like “Fully trained MAML” etc.), and the statements about comparisons (e.g. in intro and conclusions) should qualify that MetODS performs better “[for adapting a single layer of weights] at the beginning of training.”
* MetODS consists of a complicated set of changes to the architecture; some ablation experiments should be performed to demonstrate the contributions of different changes. A few such changes are tested for Harlow (S=1 and no element-wise weights), but it would be ideal to see these on a few other tasks, as well as conditions like the following:
  - It wasn’t entirely clear whether beta and kappa are element-wise parameters or scalar; assuming the latter, switching to the scalar version would be a useful ablation.
  - What if the authors kept all aspects of MetODS but switched from the Hebbian update rule to an alternative rule. Ideally this would include something like a hypernetwork (see below), but if that is too computationally expensive there are a variety of changes that could be explored, e.g. instead of the updating based on the outer product of v with itself, what if learned linear projection weights from v to two new vectors q and p and then used a (weighted) outer product of q and p? This may not be the best experiment; the point is just that more exploration of which aspects of the update rule are important (e.g. whether the Hebbian update is somehow intrinsically useful)  would help future researchers understand what to explore.
  - Adding features to the baselines could be useful too in determining where the real benefits are; e.g. if the authors used MAML with S (smaller) updates per timestep rather than 1, how does it compare? What if they augmented MAML with parameter-specific meta-learned update weighting and smoothing constants alpha and beta? Etc.
* I found the connection to the policy transport perspective somewhat unsatisfying. A substantial portion of the paper (~1.5 pages) is spent presenting this perspective, but it seems like the efficiency, capacity, and generality can be (and have been) previously defined without adopting this perspective.
  - I believe that the policy transport perspective could be removed or substantially reduced; indeed, I think that this would perhaps make the paper clearer by focusing more directly on the algorithmic contributions.
  - The three concepts defined have prior precedents in the literature; for example the notion of cumulative regret has a long history in RL, has often been used in meta-RL (e.g. Wang et al., 2016), and is effectively a measure of efficiency. The paper should connect these concepts to the relevant prior ideas.
  - If the authors include the policy perspective in the paper, it would be ideal to *compare* their approach to the baselines with respect to policy transport. For example, if they visualized policy transport for other approaches (e.g. MAML) like they do for their own approach in Fig. 2C, and then *quantified* the advantage of their approach in terms of policy transport (e.g. showing that their method takes a more direct path in policy space than the baselines), that would be much more compelling.
  - As one additional note of interest, the policy transport perspective reminded me a bit of a recent paper I saw on how fast policies update with a gradient update to the weights (Schaul et al., 2022); it seems potentially relevant to thinking about the policy transport perspective, although it is not primarily focused on optimal changes in the policy so much as noisy ones (to my understanding; I’ve not actually read Schaul et al. yet).
* The discussion of the prior literature seemed a little skewed to me in places; while it is true that few prior approaches have pursued fast weights “as a function of the current synaptic state or external reward signals” per se, there has been work on using fast weights in Meta-RL (e.g. Sarafian et al., 2021) that probably should be discussed as relevant background.

The authors do not necessarily have to run every experiment suggested above (which might be infeasible) for me to consider the paper publication worthy. However, the paper would need to would need to compare to stronger baselines (including the original MetaWorld results, if it is computationally infeasible to train their own strong baselines), would need to more clearly describe the experiments they performed, and would need to outline the limitations of the experiments and approach in more detail.

---

> ### Author Response · Authors · 2022-08-01
> **Reply to you initial review [1/n]**
>
> **Thank you for the very thorough review that you conducted on our work. We appreciate your interest and your supportive comment despite the grade and would like to engage discussion on a few points that you raised. Please see below, we refer to your review augmented with our reply.**
>
> ## 1- Unfair comparison to baselines
>
> >**We answer with additional clarifications for comment 1 regarding unfair training budget and we correct the misunderstanding regarding comment 2 about restriction of baselines.**
>
> "[...] The authors limit to 10M steps for the MetaWorld ML1 and ML10 environments; but the original MetaWorld paper achieved much higher performance with RL^2 and MAML after much longer training (300M steps). [...]"
>
> > **Indeed, for ML1 and ML10 in the MetaWorld experiment, we are testing all of our models on a limited but fair-to-all computation budget of 10M steps, using the official benchmark pipeline and parameters provided by the _Garage_ library [1]. The reason we set this particular experiment to 10M steps is because it represents an already large amount of compute and time (each run for any method takes approximately a week for reaching 10M steps despite parallelisation on a 10 cpu cluster) as well as a large amount of interaction with the environnement (20K episodes and ~1000 PPO iterations). While we agree that baselines have not converged to the final 300M steps performance reported in the Meta-World paper benchmark, nor has METODS, as we still witness increases in test performance at 10M steps. We add that success rate curves can be noisy (crossing curves) but test performance remains above what is reported for RL2 and MAML on ML10. To convince you more, we ran our model for an extra 5M steps on ML1-push and ML10 and found that MetODS is still ahead in terms of test performance (Push at 0.33% and ML-10 at 0.19% at 15M steps which is already the asymptotic level of test performance for MAML and RL2 at 300M steps).**
>
> > **In all other experiments, notably the Mujoco robot control and the maze navigation experiments), we trained models to full convergence (at 1e7 timesteps) and in these cases, MetODS consistently overperforms baselines early-on in training and achieves overall better performance. Hence, we considered that this budget, given computation constraints, was sufficient to demonstrate the potential of our synaptic reinforcement learning rule.**
>
> >**We agree with your commentary regarding the necessity of mentionning the official benchmark of MetaWorld and added these final baseline results to our experimental section for comparison purpose.**
>
> "[...] The authors limit their approach to a single layer of adaptable weights [...] If the baselines are restricted compared to the original work this limitation would be a major caveat that should be stated much more explicitly in the text. [...]"
>
> > **Regarding the baselines, we strictly follow the original implementation of authors and do not restrict their adaptation in any way. For MetaWorld, we run the official baselines from Garage with the exact provided setting and note that baselines performance closely follows reported performance metrics on this budget. For the Maze, Mujoco and Harlow experiments, we strictly follow [2] for MAML for model architecture with a three layer perceptron of 100 hidden units with ReLU non-linearities and use gradient descent over the whole network. We tried to tune the inner learning-rate in the Maze experiment with not much difference in overall performance (reported lr=0.05). For RL2 [3], we used a GRU cell with 100 hidden units and tank non-linearity that has the same number of parameters than MetODS, hence matching model complexity.  In both cases, performance closely follow metrics already presented in other works. We regret this misunderstanding and we clarify this by adding a specific section for describing baselines in the main text.**
>
> "[...] Unless I’m misunderstanding both the above points, I find statements like “producing better agents than previous meta-RL approaches” [...] to be somewhat misleading.[...]"
>
> > **While we reaffirm that we tried to give the fairest comparison possible to our model in every proposed experiments with proven meta-RL frameworks unmodified from original description, we agree that our goal is not to propose a definitive model for Meta-RL, but rather to inspire researchers with original computational principles such as the presented meta-learned hebbian plasticity and recursive updates.**
>
> >**To your point, we mitigate these claims in the second version to better emphasise our explanatory perspective instead. Again, all experiments apart from MetaWorld are trained to full convergence and we preferred running multiple experiments over diverse domains as we believe that the diversity of tasks in which the synaptic rule performed well is a stronger demonstration of the potential of exploring recursive Hebbian updates for meta-reinforcement learning.**
>
> [1/n]

---

> > ### Author Response · Authors · 2022-08-01
> > **Reply to you initial review [2/n]**
> >
> > ## 2 - Presentation of model, ablation and variations of the update rule
> >
> > "MetODS consists of a complicated set of changes to the architecture; some ablation experiments should be performed to demonstrate the contributions of different changes. A few such changes are tested for Harlow (S=1 and no element-wise weights), but it would be ideal to see these on a few other tasks, as well as conditions like the following:"
> >
> > > **We are not sure about the architecture that you are referring to in the first sentence of this paragraph. Could you help us clarify this point? Nevertheless, the different points below are well taken and converge with other reviewers comments. Hence, we retrospectively agree that we can better clarify the model definition, specifically stating more clearly the originality of the model and the role of its different components, which we propose to do in our revised version.**
> >
> > "It wasn’t entirely clear whether beta and kappa are element-wise parameters or scalar; assuming the latter, switching to the scalar version would be a useful ablation."
> >
> > > **Beta and Kappa are scalar parameters and not matrices, but they are differentially tuned with respect to the iteration of the recursive read-write operations. (i.e K^(l)_s refers, at iteration s, to the contribution of the pattern in the previous iteration l.) We will add clarifications to the model description in section 3 line 160.) These parameters could be compared to temporary interacting neuro-chemical constants driving changes in signal transduction in biological synapses. In this sense, they were thought to be shared at the neuron population level. But indeed, there could be interest in defining synapse-wise parametrisation of such constants, at the expense of a substantial parameter increase in O(N^2) with N the numbers of neurons in the plastic layer.**
> >
> > What if the authors kept all aspects of MetODS but switched from the Hebbian update rule to an alternative rule. Ideally this would include something like a hypernetwork (see below), but if that is too computationally expensive there are a variety of changes that could be explored, e.g. instead of the updating based on the outer product of v with itself, what if learned linear projection weights from v to two new vectors q and p and then used a (weighted) outer product of q and p? This may not be the best experiment; the point is just that more exploration of which aspects of the update rule are important (e.g. whether the Hebbian update is somehow intrinsically useful) would help future researchers understand what to explore.
> >
> > > **This is a very interesting perspective for developing Meta-RL programs as we believe in the general idea of learning parametric learning rule, be they Hebbian or driven by another neural networks. MetODS is a demonstration of the former and has the advantage of a biological grounding and lean parametrisation, but we agree that the advantage of this framework is also the flexibility in the definition of the read-write functions that calls for more exploration.**
> >
> > > **In this sense, we actually tested several variations for the maze experiment, namely the outer-product linear key-query projections that you are hinting at and will gladly incorporate these results to better build insights on the inner working of the recursive scheme. We also find the hypernetworks idea interesting as we think that non-linearity could potentially update weights in a more complex way, although define non-linear meta-networks did not yield improvement in our exploration and changed the nature of update that we explore in this work.**
> >
> > "Adding features to the baselines could be useful too in determining where the real benefits are; e.g. if the authors used MAML with S (smaller) updates per timestep rather than 1, how does it compare? What if they augmented MAML with parameter-specific meta-learned update weighting and smoothing constants alpha and beta? Etc."
> >
> >  > **We agree that densifying the baselines experiments could help better seize the benefits of this particular update rule, however we note that proposing S smaller updates of MAML per time-step is not a suitable experiment since the gradient estimate is a function of the current weight state: Either that would require sampling again the environment for producing a new gradient estimate which boils down to the single update scheme, or that would require to use the previous gradient estimate several times, which will also amount to the original single update. Instead, we propose to integrate the experiment on adding synapse-wise tuning of plasticity in MAML and note that contrary to MetODS, this supplementary parametrization is adversarial to performance.  Additionally, we note that MAML performance in this experiment is nowhere near the one obtained through continual update mechanisms (MetODS and RL^2).**

---

> > > ### Author Response · Authors · 2022-08-01
> > > **Reply to you initial review [3/n]**
> > >
> > > ## 3 - Theoretical introduction and connection to previous literature
> > >
> > > "I found the connection to the policy transport perspective somewhat unsatisfying. A substantial portion of the paper (~1.5 pages) is spent presenting this perspective, but it seems like the efficiency, capacity, and generality can be (and have been) previously defined without adopting this perspective. I believe that the policy transport perspective could be removed or substantially reduced; indeed, I think that this would perhaps make the paper clearer by focusing more directly on the algorithmic contributions."
> > >
> > > > **Although we considered this a necessary discussion to introduce the different aspects tested for our meta-RL adaptation mechanism, we retrospectively agree with your remark and worked on condensing this expended interpretation, in order to leave room for an improved model and experiments description as well as ablations mentioned above.**
> > >
> > > "The three concepts defined have prior precedents in the literature; for example the notion of cumulative regret has a long history in RL, has often been used in meta-RL (e.g. Wang et al., 2016), and is effectively a measure of efficiency. The paper should connect these concepts to the relevant prior ideas."
> > >
> > > > **Agreed. In accordance with your previous point, we propose to ground better our description of the considered aspects of meta-RL (efficiency, capacity and generality) in previous literature by condensing the policy transport section.**
> > >
> > > "If the authors include the policy perspective in the paper, it would be ideal to compare their approach to the baselines with respect to policy transport. For example, if they visualized policy transport for other approaches (e.g. MAML) like they do for their own approach in Fig. 2C, and then quantified the advantage of their approach in terms of policy transport (e.g. showing that their method takes a more direct path in policy space than the baselines), that would be much more compelling."
> > >
> > > > **Indeed, we agree that this comparison would be beneficial to connect our perspective on policy transport with the different particularities of baselines and give a qualitative comparison of the benefit of the update rule. We are currently trying to align the policy space visualisation with respect to the different adaptation mechanisms at the moment and we will eventually try to add a full comparison with MAML in weight space as well as RL^2 in activation space in section 5.1**
> > >
> > > "As one additional note of interest, the policy transport perspective reminded me a bit of a recent paper I saw on how fast policies update with a gradient update to the weights (Schaul et al., 2022); it seems potentially relevant to thinking about the policy transport perspective, although it is not primarily focused on optimal changes in the policy so much as noisy ones (to my understanding; I’ve not actually read Schaul et al. yet)."
> > >
> > > > **We postulate that you refer to [2]. Thank you for raising this recent reference. Indeed, quantifying the effect of rapid weights updates on a neural network policy changes is crucial for RL at large and we believe that our original update rule can bring additional insight on the sensibility of a neural network policy to its weight state. For instance, in the Harlow one-shot learning experiment, we interpret weight updates as a drastic change in Hopfield energy of the fast weights, which in turn alter the network response to future stimuli. We plan to explore more this perspective in future work, as well as exploring the notion of stochasticity in fast weight updates as a substrate for curiosity during policy adaptation.**
> > >
> > > "The discussion of the prior literature seemed a little skewed to me in places; while it is true that few prior approaches have pursued fast weights “as a function of the current synaptic state or external reward signals” per se, there has been work on using fast weights in Meta-RL (e.g. Sarafian et al., 2021) that probably should be discussed as relevant background."
> > >
> > > > **Thank you for sharing this very relevant work, of which we must admit that we were not aware. Hypernetworks for Meta- or Multi-task RL is an interesting proposal for learning non-linear mapping from context to weight parametrisation that can improve over vanilla gradient update and we will definitely include this line of work in our discussion.**

---

> > > > ### Author Response · Authors · 2022-08-01
> > > > **Reply to your review [4/n] and synthesis**
> > > >
> > > > "The authors do not necessarily have to run every experiment suggested above (which might be infeasible) for me to consider the paper publication worthy. However, the paper would need to would need to compare to stronger baselines (including the original MetaWorld results, if it is computationally infeasible to train their own strong baselines), would need to more clearly describe the experiments they performed, and would need to outline the limitations of the experiments and approach in more detail."
> > > >
> > > > > **We take good note of this synthesis and present along with this reply, a novel version of our submission. Specifically, we sum up the main modifications in relation to your comments:**
> > > >
> > > > **1 ) We augmented our section 5 (Experiments) with a more comprehensive description of our experiments as well as additional results on MetODS:**
> > > >
> > > > - We add a more thorough description of the baselines used in all experiments used to compare with our model.
> > > > - We add full results reported in [1] for the Meta-World experiment for comparison with our restrained training budget.
> > > >
> > > > **2 ) We better motivated and clarified our synaptic meta-learned rule in section 3 (MetODS)**
> > > >
> > > > - We reworked our model introduction by better motivating the components of our weight update rule in relation with experimental results and ablations.
> > > > - We integrated to the maze experiment a suggested ablation study as well as a discussion on variations of the update rule of MetODS.
> > > > - We also add additional results for baselines, notably augmenting MAML with element-wise synaptic tuning parameters and on the policy transport perspective in the Harlow experiment.**
> > > >
> > > >
> > > >  **3 ) To accommodate for these changes and better connect our discussion with previous litterature, we reworked our theoretical discussion on the policy transport perspective in section 2 (Background)**
> > > >
> > > > We connected our theoretical perspective with previous concepts of Reinforcement Learning (namely, cumulative regret, RL as bayesian task inference…)  in section 2 and mention more recent work on fast weights for Meta-RL such as [5].
> > > >
> > > > **We thank you again for the many insightful comments as we think they contribute to strenghten overall our submission. As you seem supportive of this work, we hope that such changes could help you reconsider your grade and are happy to discuss further with you these different points.**
> > > >
> > > > [1] Garage: A toolkit for reproducible reinforcement learning research, 2019, The garage contributors
> > > >
> > > > [2] Model-Agnostic Meta-Learning for Fast Adaptation of Deep Networks
> > > > Chelsea Finn, Pieter Abbeel, Sergey Levine Proceedings of the 34th International Conference on Machine Learning
> > > >
> > > > [3] RL2: Fast Reinforcement Learning via Slow Reinforcement Learning Yan Duan, John Schulman, Xi Chen, Peter L. Bartlett, Ilya Sutskever, Pieter Abbeel
> > > >
> > > > [4] The Phenomenon of Policy Churn, Tom Schaul, André Barreto, John Quan, Georg Ostrovski, Preprint
> > > >
> > > > [5] Recomposing the Reinforcement Learning Building Blocks with Hypernetworks Elad Sarafian Shai Keynan Sarit Kraus

---

> > > > > ### Comment · Reviewer_aCER · 2022-08-03
> > > > > **Thanks; still some lingering concerns, but the paper seems closer to acceptable.**
> > > > >
> > > > > Thanks to the authors for their thorough response, and apologies for failing to include the references list with my original review—I have attached it below, though it sounds like the authors figured out which papers I meant. I do feel like the paper has improved, and more clearly states what is conveyed by the experiments. I still feel that it would be better to run a full comparison to the MetaWorld baselines (and while the authors are right that their approach is on par with 20% test performance on ML10, where every method does relatively poorly, 33% is comparable to asymptotic performance for ML1-Push), but I understand that it may be infeasible to do so. I will update my score accordingly.
> > > > >
> > > > >
> > > > >
> > > > > References
> > > > > -------
> > > > >
> > > > > Sarafian et al., 2021: http://proceedings.mlr.press/v139/sarafian21a.html
> > > > >
> > > > > Schaul et al., 2022: https://arxiv.org/abs/2206.00730
> > > > >
> > > > > Wang et al., 2016: https://arxiv.org/abs/1611.05763

---

### Official Review · Reviewer_dG2z · 2022-07-10

**Rating:** 8
**Confidence:** 3
**Soundness:** 4 excellent
**Presentation:** 3 good
**Contribution:** 4 excellent

**Summary:**

The authors propose MetODS: Meta Optimized Dynamical Synapses, a meta-RL algorithm that learns by updating its weights through interaction with the environment and its own current weight state. The update rule is recursively applied and allow the algorithm to learn relations between stored patterns and incoming information. MetODS notably uses an element-wise weighting, that allows for different plasticity amplitudes at every connection. The authors then experimentally compare their algorithm to RL2 and MAML and analyze the performances of their algorithms against these baselines in terms of efficiency (One-shot learning and rapid motor control), capacity (achievable level of performance for a distribution of task) and generality (how well the policy transfers to tasks unseen during training).

**Questions:**

What is the (high level) intuition behind the "read" and "write" operation of the update rule ?

writing suggestion/typos:
l 99 and l 296: broken citation
l 139: Synapses (with capital S)
l 174: s instead of S
l 290: "to" is repeated

**Limitations:**

I haven't seen any discussion on potential negative societal impact. I, however, cannot really find non-generic remarks on this topic for this work.

**Strengths And Weaknesses:**

The paper is well motivated by biological and existing artificial neural methods. It clearly presents a novel meta RL approach and compares its performances to Meta RL baselines (namely RL2 and MAML). While the claim on efficiency and generality seems to me well-supported by the experimental evaluation, I have difficulties to understand how the evaluation on the randomly generated grid world helps to evaluate the capacity of the learner. Are different generation patters used between training and evaluation?
Otherwise, I think that the authors could sometimes add high level motivational details on the design choices of the algorithm (notably on the multi-step scheme).

---

> ### Author Response · Authors · 2022-08-01
> **Thank you for your interest and positive feedback!**
>
> **Thank you for this positive review! This is sincerely motivating and we are glad that our work convinced you. We answer more specifically your comments below:**
>
> "The paper is well motivated by biological and existing artificial neural methods. It clearly presents a novel meta RL approach and compares its performances to Meta RL baselines (namely RL2 and MAML). While the claim on efficiency and generality seems to me well-supported by the experimental evaluation, I have difficulties to understand how the evaluation on the randomly generated grid world helps to evaluate the capacity of the learner. Are different generation patters used between training and evaluation? Otherwise, I think that the authors could sometimes add high level motivational details on the design choices of the algorithm (notably on the multi-step scheme)."
>
>  > **Here we refer to capacity of a learning algorithm as the level of adaptation  that an agent can achieve when exposed to a given task. In the maze experiment, we test our meta-learners on a batch of 1000 original generated mazes that the learners have not seen during training. Since the mazes are all different with a much higher algorithmic complexity than Harlow or motor control experiments, it directly tests the learner ability to tune its policy to the precise structure of these instances (here the maze configurations). We believe that this a good test of a learner capacity to adapt its policy because it requires to dynamically articulate diverse pieces of experience into a coherent and efficient policy that match the precise maze configuration (for instance “after encountering such corner, that follows such corridor, I know that it is where the target is…” etc)**
>
> "[...] Otherwise, I think that the authors could sometimes add high level motivational details on the design choices of the algorithm (notably on the multi-step scheme).
> Question: What is the (high level) intuition behind the "read" and "write" operation of the update rule ?"
>
> > **We duly note this comment and worked on expanding the explanatory section of the model in a second version of the submission that we push along with this reply. The read and write operations are motivated by biological synaptic computation: Writing consists in an outer product that emulates a local Hebbian rule between neurons, while reading correspond to the non-linear response of the neuron population to a specific activation pattern. (\sigma(Av) = v’).**
>
> > **While a single iteration of these two operations can only add external information into weights (for writing) and retrieve a similar pattern (as reading consist in a Hopfield update), augmenting the system with recursive iterations offers a much more potent computational mechanism to filter external information with respect to the current weight state: The final activation pattern now becomes a non-linear mix of the incoming impulse v(0) and previous stored patterns which are presumably relevant to inform the agent policy, while for writing, it can also update or reinforce previous belief stored in the weights.**
>
> >**Additionally, differentially tuning the influence of previous activation patterns in the recursion through parameters kappas and betas allows to potentially emulates complex cascades of temporal modulation mechanisms found in biological synapses.**
>
> "writing suggestion/typos: l 99 and l 296: broken citation l 139: Synapses (with capital S) l 174: s instead of S l 290: "to" is repeated"
>
> > **Thanks for the raising typos, of course they will corrected in the updated submission version. However, note that we refer indeed to capital S in line 174.**
>
> "I haven't seen any discussion on potential negative societal impact. I, however, cannot really find non-generic remarks on this topic for this work."
>
> > **Indeed, at this point, we thought that our work was too exploratory to be directly exposed to negative societal impact. However, we generally believe that the potential of meta RL to automatically reveal new learning strategies must come with precaution regarding the risks to introduce new uncontrolled biases in machine learning.**

---

> > ### Comment · Reviewer_dG2z · 2022-08-06
> > **Thank you for your clarifications**
> >
> > Thank you for your clarifications.
> >
> > When I asked "What is the (high level) intuition behind the "read" and "write" operations of the update rule ?", I meant a higher level one. As you said, this rule isn't commonly used in the field yet. Maybe you could provide a concrete example of a situation (in which an agent is having a suboptimal behaviour), detailing the high level computations of these two operations.
> >
> > I think it could help readers that are not familiar with this rule to grasp how it works, and its difference to GD.

---

### Author Response · Authors · 2022-08-02
**Response to all reviewers**

We thank all reviewers for their time and interest in reviewing this paper, as well as for the helpful comments that help us strengthening our submission. Along with detailed responses below, we uploaded an updated version addressing issues raised by reviewers and are actively working on a definitive submission proposal in line with all comments.

 - Specifically, **we focused on improving definitions and motivations of the computational mechanisms supporting MetODS learning** as we identified this comment to be shared by reviewer dG2z, aCER and 7zkD:
	- We improved model introduction in section 3 to better motivate the exploration of fast weight for meta-RL with respect to previous theoretical discussion (section 2) on policy transport. (aCER)
	- We detailed better the computational principles on which MetODS is based (local tuning, recursive updates, read-write mechanism…) emphasising their originality and grounding in neuroscience. (dG2z, 7zkD). We additionally reworked figure 1 to better serve our model introduction.
	- We justified the importance of each computational component by running an ablation study in the maze experiment (section 5.2) as well as proposing experimental variations of the writing mechanism. (aCER)
 - Additionally, we noted that some **clarifications were needed regarding baselines settings** and we proposed to add a detailed description in S.I. (aCER, 7zkD)

Lastly, we want to re-emphasize that one major contribution of our work is, in our opinion, to demonstrate that self-contained learning program can emerge spontaneously from sheer optimization of the right class of synaptic control models (here based on neuroscience principles) and can depart strongly from classical gradient-based algorithms. As machine learning is evolving more and more towards automatic discovery of computational components versus engineered ones, we believe that this works is a natural step in this direction and can inspire researchers towards automatic discovery of new learning algorithms.

---

### Author Response · Authors · 2022-08-08
**Second response to reviewers**

We thank reviewers for their diligence in evaluating our modifications and willingness to increase their score. We are enthusiastic about our work forming a stronger contribution thanks to their feedback!

Regarding last comments from reviewers, we are currently working on delivering an additional analytical comparison (rev. 7zkD) between MetODS and MAML synaptic updates in the Harlow task (section 5.1) that will connect better our perspective on policy transport (rev. aCER) and provide additional justification for our dynamic Hebbian update compared to gradient-based approaches (rev. dG2z), that we will integrate to a definitive version in the coming days.

---

### Meta-Review · Area_Chair_wg8t · 2022-08-25

**Recommendation:** Accept
**Confidence:** Certain

**Metareview:**

This is exciting work that demonstrates the ability of self-modifying networks to solve meta-reinforcement learning problems. The reviewers all agree that this is strong work, and the authors have convincingly addressed most of the concerns the reviewers brought up during the reviewing phase. There are a few lingering questions about the applicability of the baselines, but these are quite minor. The authors have further promised to add analytical comparisons and additional details /motivation on the Hebbian update. Given this, I view this paper quite positively and encourage the authors to integrate the additional experiments and details they mentioned in the feedback stage.

**Award:**

No

---

### Decision · Program_Chairs · 2022-09-14

Accept